# A Concentration Gradients Tunable Generator with Adjustable Position of the Acoustically Oscillating Bubbles

**DOI:** 10.3390/mi11090827

**Published:** 2020-08-31

**Authors:** Bendong Liu, Zhigao Ma, Jiahui Yang, Guohua Gao, Haibin Liu

**Affiliations:** 1Faculty of Materials and Manufacturing, Beijing University of Technology, Beijing 100124, China; mzhigao_ny@163.com (Z.M.); klrs80@sohu.com (J.Y.); gaoguohua@bjut.edu.cn (G.G.); liuhb@bjut.edu.cn (H.L.); 2Electrical and Mechanical College, Beijing Vocational College of Agriculture, Beijing 102208, China

**Keywords:** concentration gradients, acoustically oscillating bubbles, polydimethylsiloxane (PDMS) wall, temporal controllability

## Abstract

It is essential to control concentration gradients at specific locations for many biochemical experiments. This paper proposes a tunable concentration gradient generator actuated by acoustically oscillating bubbles trapped in the bubble channels using a controllable position based on the gas permeability of polydimethylsiloxane (PDMS). The gradient generator consists of a glass substrate, a PDMS chip, and a piezoelectric transducer. When the trapped bubbles are activated by acoustic waves, the solution near the gas–liquid interface is mixed. The volume of the bubbles and the position of the gas–liquid interface are regulated through the permeability of the PDMS wall. The tunable concentration gradient can be realized by changing the numbers and positions of the bubbles that enable the mixing of fluids in the main channel, and the amplitude of the applied voltage. This new device is easy to fabricate, responsive, and biocompatible, and therefore has great application prospects. In particular, it is suitable for biological research with high requirements for temporal controllability.

## 1. Introduction

Investigating the response of cells in organisms to different chemical stimuli can help demonstrate specific factors that regulate the cell biological behavior [1]. However, completing the cell experiment directly in vivo is expensive and difficult to operate, so it is necessary to create a suitable vitro environment to meet the needs of cell research. At a fundamental level, the cells in living organisms are surrounded by micro-environments of different chemical concentration gradients, which play an important role in biological phenomena [2]. Therefore, the creation of a micro-environment using a chemical gradient is a basic requirement for performing in some vitro experiments with cells, especially for analyzing the cell response of migration [3,4,5], growth [6,7], chemotaxis [8,9,10], and differentiation [11,12] under a real-time adjustable chemical gradient. In addition, it is also adopted for completing the screening and toxicity detection of drugs [13,14,15,16], based on the cellar activity and behavior in the dynamic gradients environment. Several gradient generators used to build specific concentration gradients has been demonstrated. Many of these generators mix solutions based on convection and diffusion principles, including tree-shaped structure [17,18,19], Y-shaped structure [20,21], membrane system [22,23], droplet-based system [24,25], hydrodynamic system [26,27], and microarray [28,29]. Most of these passive devices have difficulty regulating the gradients in the channel spatiotemporally; they are only suitable for experiments that require the creation of a stable concentration gradient micro-environment.

As biochemical gradients in vivo may never be static, investigations into their cellular response to gradients require experimental means in order to control the gradients in a dynamic manner so as to reproduce a natural cellular environment [30]. Therefore, it is necessary to research gradient generators that can create dynamic concentration gradients. Recently, active gradient generators based on acoustic-driven solution mixing have attracted attention because of their operability, fast response, and non-invasion properties, and the principle of mixing liquids is to generate a microstreaming phenomenon [31] in a microchannel. Daniel Ahmed [32] presented a ladder-arranged horseshoe-shaped structure to capture the bubbles, which were driven by acoustic waves to mix the surrounding solutions and to create concentration gradients. They regulated the concentration gradients spatially and temporally by changing the applied voltage. Ghulam Destgeer [33] demonstrated a streaming phenomenon caused by focusing on the traveling surface acoustic waves acting on the fluid in a microchannel, which makes the solution mix and produce chaotic acoustic concentration gradients in a channel. Po-Hsun Huang [5] developed a method of driving sharp structures using acoustic waves to mix the solution, and produced a concentration gradient, spatially regulating the gradient by changing the driving voltage and flow rate.

Here, we propose a tunable chemical gradient generator actuated with acoustically oscillating bubbles trapped in the bubble channels in a controllable position based on the gas permeability of polydimethylsiloxane (PDMS). When the trapped bubbles are activated by acoustic waves, the solution near the gas–liquid interface is mixed. The volume of the bubbles and the position of the gas–liquid interface are regulated with the permeability of the PDMS wall. The tunable chemical gradient can be realized by changing the numbers and positions of the bubbles that enable the mixing of fluids in the main channel, as well as the amplitude of the applied voltage. Most of the previous concentration gradient generators driven by acoustic waves can only change the concentration gradient of the solution with time by switching the power supply on and off, and cannot keep a certain concentration gradient for a period of time. This new type of concentration gradient generator can make the concentration gradient of the solution change smoothly with time by moving bubbles, and can control the size of bubbles and the position of the gas–liquid interface. Therefore, by adjusting the position of gas–liquid interface, any concentration gradient generated during the change in bubble position can be maintained for a period of time. This new concentration gradient generator has better controllability. The working principle of our device is utilizing the acoustically oscillating bubble to mix the solution. The biocompatibility of the acoustic device was verified by Daniel Ahmed [34] and Yuliang Xie [35] through experiments. The solution flow rate, excitation frequency, and the applied voltage adopted in their experiments are similar to our concentration gradient generator. Therefore, this concentration gradient generator is also biocompatible. Our devices are easy to fabricate, responsive, and biocompatible; therefore, they have great application prospects. It is mainly suitable for biological research with high requirements for temporal controllability.

## 2. Theory and Device Design

As is shown in Figure 1a, the gradient generator consists of a glass substrate, a PDMS chip, and a piezoelectric transducer. The PDMS chip is designed with two inlets, an outlet, three air ports, three air channels, three bubble channels, and three crescent shaped chambers. Each of the crescent shaped chambers consist of a circular chamber, a semi-circular PDMS wall, and a sector ring chamber. The characteristic size of the gradient generator is shown in Figure 1b—the depth of all of the channels is 110 μm, the width of the main channel is 600 μm, the widths of the bubble channel and the air channel are 100 μm, and the angle between the main channel and the upper bubble channel is 120°. The distance between the upper side of the left bubble channel and the upper side of the main channel is 250 μm, and the horizontal distance between the right end of the left bubble channel and the junction of the upper bubble channel and main channel is 550 μm. The design of the micro channel and the acoustic actuator was based on our previous study on an acoustic device [36], and on the report of Daniel Ahmed [32]. Because of the permeability of PDMS, if the distance between the bubble channels and air channels is too short, there will be a pressure difference between the channels when the external pressure is introduced into some of the channels, and the gas will penetrate into other channels, which will affect the stability and controllability of the bubble movement. In order to control the position and the volume of the bubbles, crescent shaped PDMS chambers were designed between the air channels and the bubble channels. The air pressure could be stably introduced into the bubble channel through the permeability of the PDMS.

The working principle of the spatiotemporally tunable gradient generator is shown in Figure 2. When two different concentrations of solution were injected into two inlet ports, separately, laminar flow occurred in the main channel. Bubbles were generated when the fluid passed through the intersection of the bubble channels and the main channel, owing to the surface tension of the air and liquid. As the piezoelectric transducer was driven, acoustically oscillated bubbles generated am acoustic streaming effect around the air–liquid interface, and mixing was generated when the stratification lines of two different concentrations of solutions reached the region where bubbles generated acoustic streaming (Figure 2a). The acoustically oscillated hemispherical bubble could produce dramatic acoustic streaming at its natural frequency, and the frequency was estimated using the Rayleigh–Plesset equation [36].
(1)$$f^{2}=\frac{1}{4\rho \pi^{2}a^{2}}\left\{3k(p_{0}+\frac{2\sigma }{a})-\frac{2\sigma }{a}\right\}$$
where ρ is the density of the fluid (1000 kg/m^3^), σ is the surface tension of water (0.0728 N/m), k is the polytropic exponent for an air pocket (1.4), $p_{0}$ is the hydrostatic liquid pressure (101.325 kPa), and a  is the radius of the bubble at an equilibrium state (55 μm). The theoretical value of the resonance frequency (60 kHz) of the bubble with a radius of 55 μm can be obtained with Equation (1).

We introduced a semi-circular PDMS wall structure between the air channel and the bubble channel to stabilize the bubble movement. Based on the permeability of the PDMS wall, the steady-state gas flux was calculated with Equation (2) [37].
(2)$$\frac{dV}{dt}=\frac{PA\Delta p}{\delta }\frac{T}{273}\frac{76}{p_{atm}}$$
where $\frac{dV}{dt}$ is the volumetric flow rate, P is the permeability coefficient of PDMS wall for air, A is the diffusion area, Δp is the pressure difference on both side of the PDMS wall, T is the absolute temperature in kelvin, $p_{atm}$ is the atmospheric pressure, and δ is the thickness of the PDMS wall. In order to make the moving speed of the gas–liquid interface about 10 μm/s in the experiment, we designed the structure of the PDMS wall based on Equation (2). The pressure difference used in this experiment was 30 kPa, the penetration area of the PDMS wall in the semicircular chamber was 0.2 mm^2^, and the penetration length was 200 μm.

When a negative pressure source was connected with air-port 1, the bubble trapped in the upper bubble channel moved backwards along the bubble channel. If the backward distance was long enough, the oscillating bubble stopped mixing the solutions in the main channel. Then, the atmosphere pressure was connected with air-port 1 and the position of air–liquid interface was stabilized owing to the same pressure obtained on both sides of the PDMS wall. For this situation, the concentration gradients are shown in Figure 2b. When a positive pressure source was connected with air-port 1, the bubble trapped in the upper bubble channel moved forward along the bubble channel. Similarly, the tunable gradient generator generated concentration gradients as shown in Figure 2c,d, by adjusting the positions of the bubbles in the bubble channel, respectively. As a result of this, the concentration gradients of the solution were more controllable in space by adjusting the mixing of specific positions.

## 3. Fabrication and Experimental Setup

The concentration gradients generator was fabricated using standard soft lithography techniques. Briefly, a negative master was fabricated on a wafer with an SU-8 photoresist (SU-8 2050, Micro-Chem Corp., Newton, MA, USA) using a transparency mask and a mask aligner (BGT-3B, Beijing Chuangweina Technology Co. Ltd., Beijing, China). A prepolymer of PDMS (Sylgard 184, Dow Corning) and a curing agent were mixed at a ratio of 10:1 (*wt*/*wt*). The PDMS mixture was degassed in a vacuum chamber for 30 min, after which it was poured into the SU-8 master and baked on a hot plate at 65 °C for 2 h. Then, the PDMS chip was peeled from the mold, and five holes with a diameter of 1.5 mm were punched for connection with the tubes. The PDMS chip and the glass substrate were irreversibly bonded after being treated with oxygen plasma. Then, the device was placed on a hot plate at 60 °C for 15 min to make the bonding firm and stable. Finally, a piezoelectric transducer (YZ-20T-6.2A1; Yaoze Electronics) was bonded on the glass substrate near the PDMS chip with a thin layer of epoxy (Gleihow 568; Gleihow New Materials Co. Ltd., China). Figure 3 shows a photograph of the concentration gradients generator.

A schematic of the experimental setup is shown in Figure 4. Two syringe pumps (SPLab01; Shenchen, China) drove the fluid to inlet 1 and inlet 2 through Teflon tubes, respectively. The vacuum chamber and air tank provided negative pressure and positive pressure, respectively. The atmosphere, vacuum chamber, and air tank were connected with a cross four-way pneumatic joint through V3, V2, and V1, respectively. The cross four-way pneumatic joint was connected with another cross four-way pneumatic, which was connected with air-port 1, air-port 2, and air-port 3 through V4, V5, and V6 and Teflon tubes. All of the valves adopted in the experimental setup were normally closed solenoid valves (KVE21PS24N2N651A, kamoer, china). The piezoelectric transducer (YZ-20T-6.2A1; Yaoze Electronics) was actuated by a signal generator (SP1631A) with a frequency of 55 kHz. Experimentally, the resonance frequency of the intensely mixed liquid was determined by sweeping the driving frequency with a 100 Hz increment near the theoretically calculated resonance frequency (60 kHz). The main channel of the gradient generator was placed under a microscope (6XD-3, Shanghai Optical instrument, China) and was recorded with a camera and a computer.

## 4. Results and Discussion

The experimental observations demonstrate that the moving speed of the gas–liquid interface was about 10 μm/s, which is consistent with the speed obtained by theoretical calculations, and by keeping the external pressure difference constant, the gas–liquid interface moved steadily along the bubble channel at this speed. By switching the external air pressure source, the gas–liquid interface in the bubble channel moved away from or close to the main channel. Because the bubble moved slowly with a pressure difference of 30 kPa, this experiment only recorded the process of the gas–liquid interface moving away from and close to the main channel.

In order to observe the concentration gradient and the mixing effect of the solution in the main channel, deionized water and magenta ink were injected into inlet 1 and inlet 2, respectively, at the same flow rate of 3 μL/min using two syringe pumps. When V1 and V2 were turned off and V3, V4, V5, and V6 were turned on, all of the air-ports and air channels were connected with the atmosphere pressure. Three bubbles were generated as the liquid flowed through the junction of the main channel and the three bubble channels, and the liquid in the main channel was in a laminar flow. In order to study the influence of the movement of a single oscillating bubble on the mixing effect in the main channel, air-port 2 was connected with the vacuum chamber through solenoid valves. Because of the different pressure between the gas channel and the bubble channel, the gas in the lower bubble channel passed through the PDMS wall. Therefore, the gas volume of the lower bubble channel became smaller, and the gas–liquid interface moved along the bubble channel and away from the main channel. When the gas–liquid interface was about 200 μm away from the main channel, air-port 2 was connected with the atmospheric pressure to keep the gas–liquid interface unchanged. The piezoelectric transducer was turned on at a voltage of 12 V_PP_ and frequency of 55 kHz, and the oscillating bubble trapped by the left bubble channel mixed deionized water with ink around the left bubble. When the mixed liquid flowed to the mixed region of the bubble trapped by the upper bubble channel, the upper oscillating bubble diluted the mixed liquid by mixing deionized water with the ink-deionized water mixture. At this time, the lower bubble was in the lower bubble channel and did not have any mixing effect on the solution in the main channel. Then, we connected the negative pressure, atmospheric pressure, and positive pressure air sources to air port 1 individually, so that the upper bubbles were far away from, stopped, and closed the main channel in turn, and we recorded the influence of the upper oscillating bubbles on the concentration gradient in the main channel (as shown in Appendix A). Figure 5a shows the variation of the concentration gradient of the solution in the main channel caused by the displacement of the upper oscillation bubble. Figure 5b shows the variation of the concentration intensity in the marked area of Figure 5a when the bubble position was changed. The experimental results show that when the gas–liquid interface was about 40 μm away from the main channel, the oscillation of the bubble no longer mixed the solution in the main channel. At this time, the atmospheric pressure was connected to the gas channel, as result of that, the gas–liquid interface did not move any more, and the concentration intensity in the marked area did not change. When air-port 1 was connected to a positive pressure, the gas–liquid interface moved along the bubble channel to the main channel, and the oscillation of the upper bubble mixed the solution in the main channel again, and the concentration intensity of the marked area gradually increased. The moving speed of the gas–liquid interface was adjusted by the pressure of the connected gas source, or by adjusting the diffusion thickness and area of the PDMS wall. The velocity of the gas–liquid interface was about 10 μm/s when the pressure difference was 30 kPa. Figure 5c shows the variation of the mixing width of the solution as the upper oscillating bubble moves along the bubble channel. When the bubble was far away from the main channel, the mixing width decreased until it did not affect the solution mixing in the main channel. When the bubble moved to the main channel, the mixing width increased. By controlling the moving process of the bubbles on one side, the intensity and width of the concentration layer on the side could be adjusted.

We also studied the influence of the sequential movement of two bubbles on the concentration gradient of the main channel. The flow rate of the solution and the applied voltage of the transducer was the same as those for the experiment mentioned above. Three bubbles were generated as the liquid flowed through the junction of the main channel and the three bubble channels. The oscillating bubble trapped by the left bubble channel mixed deionized water with ink around the left bubble. When the mixed liquid flowed to the mixed region of bubbles trapped by the upper bubble channel and the lower bubble channel, the upper oscillating bubble diluted the mixed liquid by mixing deionized water with the ink-deionized water mixture, and the lower bubble was concentrated in the mixed liquid by mixing ink with ink-deionized water mixture. This process generated spatial concentration layers, including the dilution of the mixture of deionized water and ink, the mixture of deionized water and ink, and the concentrate of deionized water and ink (Appendix A). When the voltage was increased to 14 V_PP_ and 16 V_PP_, the mixing width around the three bubbles increased as shown in Figure 6a. Then, the piezoelectric transducer was turned off, air-port 2 was connected with the negative pressure, and the lower bubble moved backward about 40 μm along the lower bubble channel. Then, the piezoelectric transducer was turned on at voltage of 12 V_PP_, and the concentration gradient of the dilution of the ink–deionized water mixture, ink–deionized water mixture, and ink were generated with the left bubble and the upper bubble. As the applied voltage increased to 14 V_PP_ and 16 V_PP_, the mixing distance of each oscillating bubble increased the concentration gradient in the main channel, as shown in Figure 6b. Figure 6c shows the concentration gradient of the solution in the main channel mixed with the left bubble and the lower bubble with increasing applied voltage. Similarly, when air-port 1 and air-port 2 were connected to the negative pressure, the upper bubble and the lower bubble moved away from the main channel. The concentration gradient in the main channel mixed with only the left bubble with different applied voltages is shown in Figure 6d.

MATLAB was used to process the experimental image of the concentration gradient, and the gray value of the mixed solution in part of the dotted line in the main channel was extracted. The transverse variation curve of the solution concentration along the main channel was obtained by normalization processing. Three bubbles were formed at the intersection of the main channel and the three bubble channels. When the driving voltage of the sensor was 0 V_pp_, the deionized water injected by inlet 1 and the ink injected by inlet 2 were in a laminar flow state in the main channel. The concentration curve of this part was gentle on both sides and steep in the middle part. With the increasing applied voltage, the three oscillating bubbles mixed the solution around them strongly. As a result of that, the concentration curve in the main channel become smooth. When the voltage increased to 16 V_pp_, the concentration curve became an almost straight line (Figure 6e) and the solution of the whole main channel was fully mixed by the three oscillating bubbles. When the lower bubble shrank and the solution of the main channel entered the lower bubble channel longer than 40 μm, only the upper bubble and the left bubble participated in the solution mixing in the main channel. With the voltage increasing from 12 V_pp_ to 16 V_pp_, the left side of the concentration curve (i.e., the concentration of the solution at bottom of the main channel) became smooth, and the concentration layer with a higher concentration become wider (Figure 6f). This is because increasing the voltage only caused more mixing of the solution around the left bubble and the upper bubble. Similarly, the steepness of the left side of the curve and the width of the concentration layer were changed by adjusting the amplitude of the applied voltage when there were only the left bubble and the lower bubble in the main channel (Figure 6g). When there were only left bubbles mixing the solution in the main channel, adjusting the applied voltage only changed the steepness of the middle part of the curve and the width of the middle concentration layer (Figure 6h). The experimental results show that our device adjusted the concentration of the solution in the main channel in real time. The new concentration gradient generator mixed the solution with acoustic-driven bubbles, and the working principle was the same as the other mixers that adopted acoustic-driven bubbles. However, the difference from the previous studies was that the air–liquid interface and the bubbles were no longer static. The position and moving speed of the air–liquid interface and the bubbles of this device were adjustable. Therefore, by adjusting the position of the gas–liquid interface, any concentration gradient generated during the change of bubble position was be maintained for a period of time. Furthermore, by increasing the number of bubbles and reducing the radius of the bubbles, the resolution of the concentration gradient of the device was improved.

## 5. Conclusions

We present a new type of device that could tune the concentration gradient of a main channel solution in real time by moving the air–liquid interface of the oscillating bubble trapped in the bubble channel. Prototypes of the concentration generator were fabricated and some experiments were carried out. The concentration gradient in the main channel can be adjusted by moving the bubble far away from the main channel or close to the main channel. Adjustments in the intensity and width of certain concentration layers were also successfully conducted by jointly regulating the applied voltage and the number and position of the bubbles that mix the solution in the main channel. Furthermore, a semicircular PDMS wall structure was introduced between the air channel and the bubble channel, and the bubble could move smoothly with the action of the external air pressure, which made the device more operable. By adjusting the position of the gas–liquid interface, any concentration gradient generated during the change of bubble position could be maintained for a certain period of time. This new device can be applied to many biochemical studies, such as cell migration, growth, and differentiation, and drug detection, with a high real-time controllability.

## Figures and Tables

**Figure 1 micromachines-11-00827-f001:**
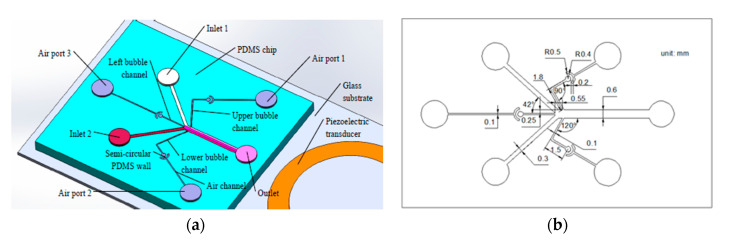
Schematic overview of the gradient generator. (**a**) Structure diagram of the gradient generator. (**b**) Characteristic dimensions of the gradient generator.

**Figure 2 micromachines-11-00827-f002:**
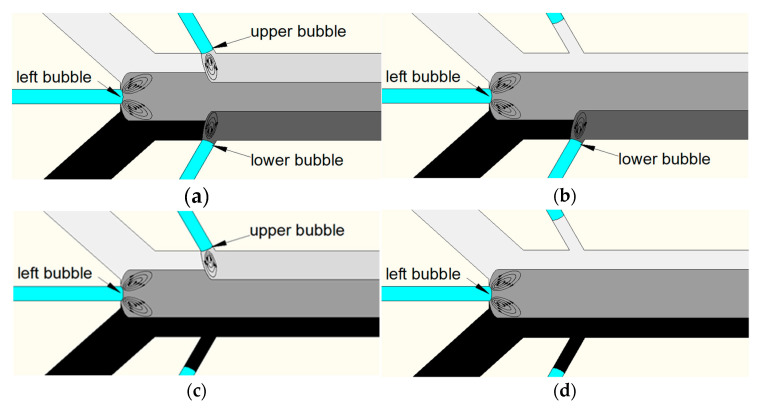
Schematic of the gradient tunable generator’s working process. (**a**) Concentration gradients with three oscillating bubbles mixing. (**b**) Concentration gradients with mixing of the left bubble and lower bubble. (**c**) Concentration gradients with mixing of the left bubble and upper bubble. (**d**) Concentration gradients with left bubble mixing.

**Figure 3 micromachines-11-00827-f003:**
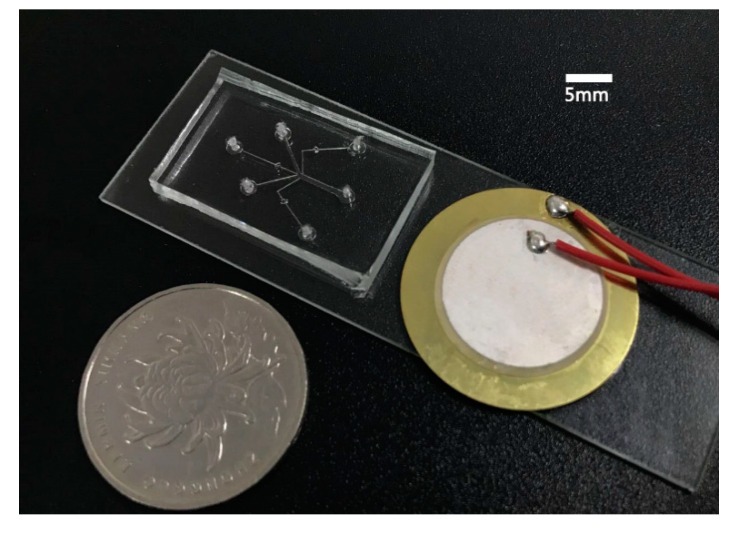
Photograph of the acoustofluidic gradient generator.

**Figure 4 micromachines-11-00827-f004:**
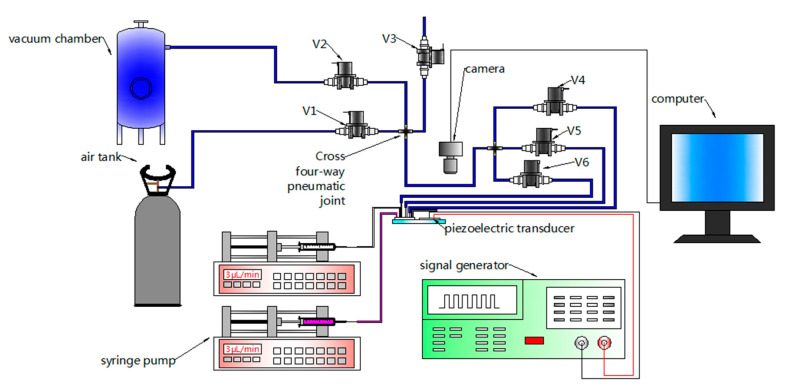
Schematic of the experimental setup for testing.

**Figure 5 micromachines-11-00827-f005:**
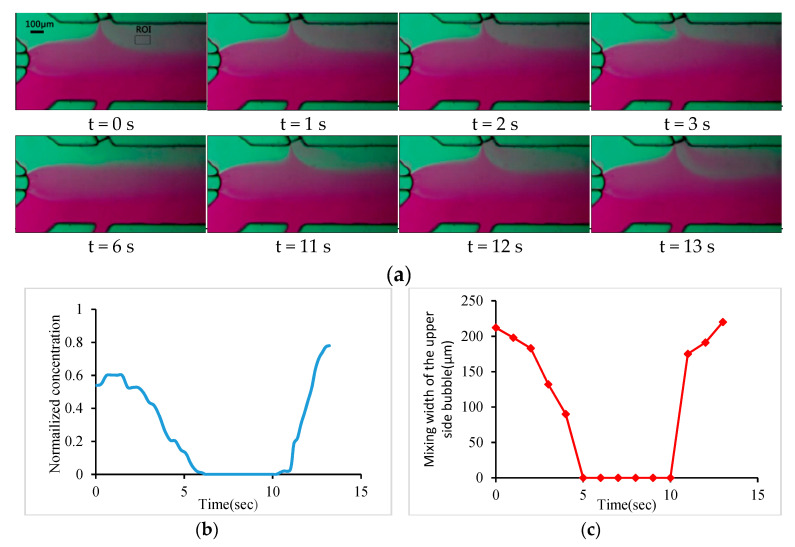
(**a**) Photographs of the concentration gradient in the main channel when the upper oscillating bubble moves away from and close to the main channel; (**b**) concentration of region of interest (ROI) during the movement of upper oscillating bubble; (**c**) mixing width of upper bubble during its movement.

**Figure 6 micromachines-11-00827-f006:**
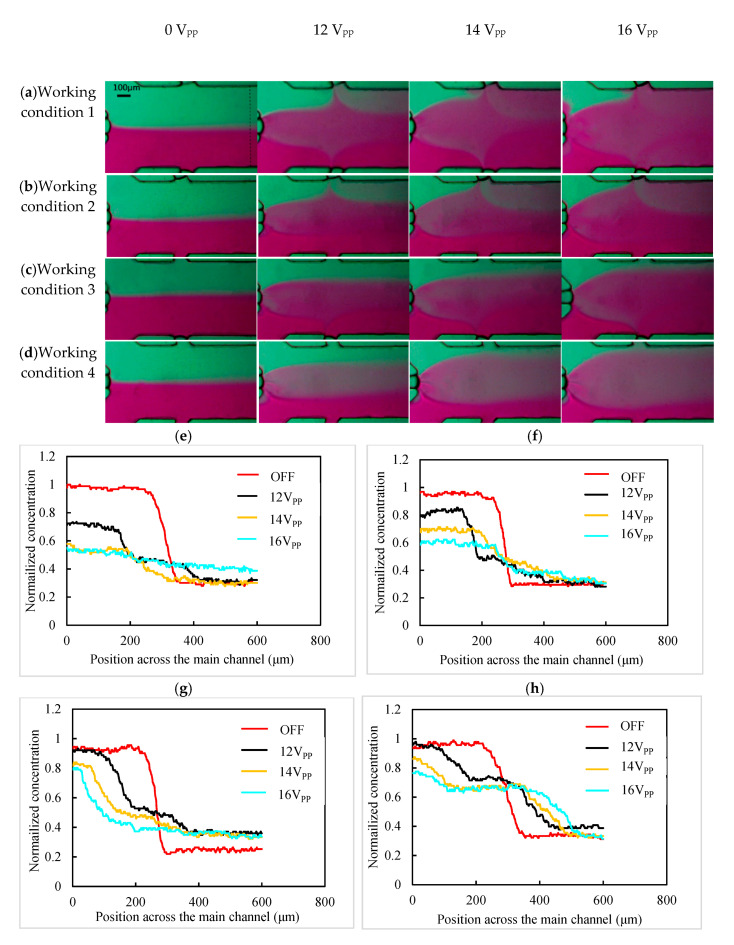
Characterization of concentration gradient in the main channel. (**a**) Images show three bubbles mixing the main channel solution. (**b**) Left and upper bubbles mixing the main channel solution. (**c**) Left and lower bubbles mixing the main channel solution. (**d**) Left bubble mixing the main channel solution under 0 V_pp_, 12 V_pp_, 14 V_pp_, and 16 V_pp_. The plots show the corresponding gradient profiles under different work conditions, namely: (**e**) work condition 1, (**f**) work condition 2, (**g**) work condition 3, and (**h**) work condition 4.

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
