# Peer review of "A Concentration Gradients Tunable Generator with Adjustable Position of the Acoustically Oscillating Bubbles"

_micromachines, 2020, doi:10.3390/mi11090827_

Round 1

Reviewer 1 Report

  1. You are referring to relatively old articles before 2015. Why are there no mentions of articles in the last five years? Have you not found any progress in this area after 2015? How did you rate the novelty of your device?
  2. How did you select the geometry of the channels and transducers, as well as the operating frequency range? Why such an arrangement of channels and transducer in space? Is your choice optimal in terms of efficiency?
  3. What about acoustic quality-factor of device and how important is it for solving specific problems in your article? Have you made any estimates of acoustic energy dissipation and its dominative mechanisms (in polymer materials, interfaces, outer boundary and volume radiation, solid-liquid and fluidic structure-transduser interactions and etc.) and how it affects the efficiency of your device?
  4. Line 216: "This new device can be applied to many biochemical studies, such as cell migration, growth, differentiation and drug detection, with high real-time controllability." What about sensitivity, selectivity and resolution? What is in comparison with analogues?

Author Response

    We would like to thank you for your constructive comments to improve our manuscript entitled “A concentration gradients tunable generator with adjustable position of the acoustically oscillating bubbles”. We really appreciate your precious time and great efforts on reviewing the manuscript. We have carefully addressed all the reviewers’ comments. All the revised portions of the manuscript are indicated with blue front. In the following sections, we answer to questions asked by reviewers.

Best regards

Bendong Liu

Comments from the editors and reviewers:

Reviewer 1

1. You are referring to relatively old articles before 2015. Why are there no mentions of articles in the last five years? Have you not found any progress in this area after 2015? How did you rate the novelty of your device?

Response 1: We are sorry for the missing the recent relative articles. And we added some references that reported in recent years according to the reviewer’s comments.

2. How did you select the geometry of the channels and transducers, as well as the operating frequency range? Why such an arrangement of channels and transducer in space? Is your choice optimal in terms of efficiency?

Response 2: The geometry design of the channels and transducers were referred to the previous research of our group[3] and the report of Daniel Ahmed[4]. We have added the explanation in the revised manuscript according to reviewer’s comments. Rayleigh–Plesset equation was utilized to calculate the theoretical resonance frequency of a bubble. Such an arrangement of channels and transducer in space is just one of our design schemes, and it needs to be optimized in the future.

[3] Bendong L , Baohua T , Xu Y , et al. Manipulation of micro-objects using acoustically oscillating bubbles based on the gas permeability of PDMS[J]. Biomicrofluidics, 2018, 12(3):034111.

[4] Ahmed D , Chan C Y , Lin S C S , et al. Tunable, pulsatile chemical gradient generation via acoustically driven oscillating bubbles[J]. Lab on a Chip, 2013, 13.

3. What about acoustic quality-factor of device and how important is it for solving specific problems in your article? Have you made any estimates of acoustic energy dissipation and its dominative mechanisms (in polymer materials, interfaces, outer boundary and volume radiation, solid-liquid and fluidic structure-transduser interactions and etc.) and how it affects the efficiency of your device?

Response 3: We are sorry for not conducting the theoretical analysis of acoustic quality-factor and acoustic energy dissipation. The purpose of this paper is just to design, fabricate a tunable concentration gradient generator and verify its feasibility. We will study the acoustic energy dissipation and the affections on the efficiency of our device in the future.

4. Line 216: "This new device can be applied to many biochemical studies, such as cell migration, growth, differentiation and drug detection, with high real-time controllability." What about sensitivity, selectivity and resolution? What is in comparison with analogues?

Response 4: The new concentration gradient generator mixed solution with acoustic-driven bubbles and the working principle is the same with other mixers that adopted acoustic-driven bubbles. However, the difference from the previous studies is that the air-liquid interface and the bubbles are no longer static. The position and moving speed of the air-liquid interface and the bubbles of this device are adjustable. By increasing the number of bubbles and reducing the radius of bubbles, the resolution of the concentration gradient of the device can be improved. We have added this discussion in the revised manuscript according to reviewer’s comments.

Reviewer 2 Report

In this paper, the authors presented a tunable concentration gradient generator by adjusting positions of oscillating interfaces between solutions and air bubbles using only one acoustic actuator. The presented concept is interesting. However, for the reasons listed below, this reviewer found it difficult how the approach might be extended by others.

Major concerns:

  1. Many tunable concentration gradient generators have been developed. The authors are suggested to prove clear significance and merits at the end of the introduction.
  2. The claim that the device is biocompatible was not supported by the data presented.
  3. The rationale for the channel design, acoustic actuator, and the excitation frequency, which appear to be critical for generating acoustic streaming-induced mixing, is not well explained. In addition, the theoretical profile of the concentration gradient wasn’t provided.
  4. The air-liquid interface shown in Fig. 5(a) (e.g., t = 6 s versus t = 11 s) only moves little and yet the concentration profile changes greatly. The authors should assess how stable and repeatable the concentration profile can be controlled. Similarly, there are no error bars for the results shown in Figs. 5 and 6.

Minor concerns:

  1. Some of the legends in Fig.6 are not shown clearly.

Author Response

We would like to thank you for your constructive comments to improve our manuscript entitled “A concentration gradients tunable generator with adjustable position of the acoustically oscillating bubbles”. We really appreciate your precious time and great efforts on reviewing the manuscript. We have carefully addressed all the reviewers’ comments. All the revised portions of the manuscript are indicated with blue front. In the following sections, we answer to questions asked by reviewers.

Best regards

Bendong Liu

Comments from the editors and reviewers:
-Reviewer 2
  1.   Many tunable concentration gradient generators have been developed. The authors are suggested to prove clear significance and merits at the end of the introduction.

Response 1: We have added significance and merits of our concentration generator at the end of the introduction according to reviewer’s suggestion. Most of the previous concentration gradient generator driven by acoustic wave can only change the concentration gradient of solution with time by switching ON and OFF the power supply, and cannot keep a certain concentration gradient for a period of time. This new type of concentration gradient generator can make the concentration gradient of solution change smoothly with time by moving bubbles, and can control the size of bubbles and the position of gas-liquid interface. Therefore, by adjusting the position of gas-liquid interface, any concentration gradient generated during the change of bubble position can be maintained for a period of time. This new concentration gradient generator has better controllability.

  1. The claim that the device is biocompatible was not supported by the data presented.

Response 2: We are sorry for our research having no related biological experiments to verify the biocompatibility of the device. The biocompatibility of the acoustic device was verified by Daniel Ahmed [1] and Yuliang Xie [2] with experiments. The solution flow rate, excitation frequency and the applied voltage adopted in their experiments are similar with our concentration gradient generator. We have added an explanation of biocompatibility of the device in the revised manuscript according to reviewer’s comments.

[1] Ahmed D , Muddana H S , Lu M , et al. Acoustofluidic Chemical Waveform Generator and Switch[J]. Analytical Chemistry, 2014, 86(23), 11803-11810.

[2] Xie, Y.L.; Nama, N.; Li, P.; Mao, Z.M.; Huang, P.H.; Zhao, C.L.; Costanzo, F.; Huang, T.J. Probing cell deformability via acoustically actuated bubbles. Small, 2016, 12, 902-910.

  1. The rationale for the channel design, acoustic actuator, and the excitation frequency, which appear to be critical for generating acoustic streaming-induced mixing, is not well explained. In addition, the theoretical profile of the concentration gradient wasn’t provided.

Response 3: The design of micro channel and the acoustic actuator was based on our previous studied on acoustic device [3] and report of Daniel Ahmed[4]. We have added the explanation in the revised manuscript according to reviewer’s comments. The excitation frequency was calculated with Rayleigh–Plesset equation. We also have added the values of the parameters in equation 1 according to reviewer’s comments. We are sorry for the lack of theoretical profile of concentration gradient. Simulation of the theoretical profile of concentration gradient is going on with software of COMSOL.

[3] Bendong L , Baohua T , Xu Y , et al. Manipulation of micro-objects using acoustically oscillating bubbles based on the gas permeability of PDMS[J]. Biomicrofluidics, 2018, 12(3):034111.

[4] Ahmed D , Chan C Y , Lin S C S , et al. Tunable, pulsatile chemical gradient generation via acoustically driven oscillating bubbles[J]. Lab on a Chip, 2013, 13.

  1. The air-liquid interface shown in Fig. 5(a) (e.g., t = 6 s versus t = 11 s) only moves little and yet the concentration profile changes greatly. The authors should assess how stable and repeatable the concentration profile can be controlled. Similarly, there are no error bars for the results shown in Figs. 5 and 6.

Response 4: We have conducted many experiments with the protypes and the experiments are repeatable. Experiments show that by adjusting the position of the gas-liquid interface and the value of the applied voltage, the concentration gradient of the solution can be adjusted. Concentration profiles in Figure 5 and Figure 6 are the observation data of one of the experiments and the data extracted directly from the experimental photos with tools of MATALB, so there is no error bar. The concentration gradient is depending on the position of the gas-liquid interface and the value of the applied voltage. In order to control the position of the gas-liquid interface and the moving speed of the air-liquid interface, a semi-circular PDMS wall structure between the air channel and the bubble channel was adopted. The speed of the air-liquid interface was about 10 μm/s in our experiments. The steady-state gas flux formula based on the permeability of the PDMS wall was added in the revised manuscript. A lower moving speed and a more precise location of the air-liquid interface can be realized by increasing the thickness of the PDMS wall and decreasing the area of the PDMS wall.
